# The Successful Use of an Ultrasound-Guided Mid-Femur Sciatic Nerve Block in a Juvenile Emu (*Dromaius novaehollandiae*) under General Anaesthesia

**DOI:** 10.3390/ani14081178

**Published:** 2024-04-14

**Authors:** Alexandru Cosmin Tutunaru, Dimitri Alarcon Morata, Valentine Pollet

**Affiliations:** Faculty of Veterinary Medicine, Clinical Department of Companion Animals, University of Liège, Av. de Cureghem 1, 4000 Liège, Belgium; dimitri.alarcon@uliege.be (D.A.M.); valentine.pollet@uliege.be (V.P.)

**Keywords:** emu, lateral luxation of the Achilles tendon, sciatic nerve block, ultrasound guided

## Abstract

**Simple Summary:**

All animals feel pain and, as vets, we have the ethical obligation to prevent it. This paper describes two anaesthetic events in an emu (*Dromaius novaehollandiae*) undergoing the surgical repair of a lateral luxation of the Achilles tendon on both hind limbs. An ultrasound-guided mid-femur sciatic nerve block was successfully performed on both occasions for intraoperative and postoperative analgesia. To our knowledge, this is the first published report of an emu who benefited from inhalation anaesthesia associated with a loco-regional block.

**Abstract:**

The current case report describes a successful ultrasound-guided mid-femur sciatic nerve block in an emu. A 2-month-old emu suffering from acute-onset lameness was referred to the University Clinic of Liège, where he was diagnosed with a lateral luxation of the Achilles tendon on both hind limbs. Two surgical procedures were performed for treatment. Both surgical procedures were performed under general anaesthesia with butorphanol, ketamine, midazolam and isoflurane in oxygen. The anaesthesia was continuously monitored. An ultrasound-guided sciatic nerve block was performed to prevent and treat surgically induced nociception. The technique was adapted from what is already described in other species. Levobupivacaine was injected perineurally under ultrasound-guidance. Intraoperative nociception was assessed based on the heart rate and mean arterial pressure changes. The recovery was uneventful and with no clinical signs of postoperative pain.

## 1. Introduction

Pain management in avian surgery is a rapidly evolving field, mirroring advancements made in mammalian regional analgesia techniques. Ultrasound and nerve stimulator-guided techniques have been previously described for application in both domestic [1,2] and wild avian populations [3,4]. Notably, extensive research conducted in canines has provided a valuable foundation for adapting nerve identification and blockade techniques for use in avian species. To our knowledge, this is the first published report of an emu who benefited from inhalation anaesthesia associated with a loco-regional analgesia technique.

Perioperative pain can induce detrimental consequences in various animal taxa, including impaired cognitive function, chronic pain states, and stress-hormone dysregulation. Therefore, we have an ethical obligation to manage nociception throughout the perioperative period and beyond in all animals, regardless of their classification as mammals or birds. A multimodal approach, including systemic analgesics and regional anaesthetic techniques, can be employed to achieve this goal. Pain recognition and assessment in birds is challenging. However, it is reasonable to believe that pain produces both sensory and emotional experiences and adverse effects, as in mammals [5]. This case report investigates the applicability of an ultrasound-guided sciatic nerve-block technique in a novel species, the emu. By demonstrating successful implementation, this report aims to contribute to the transferability of this regional anaesthetic technique across avian species.

## 2. Case Description

A 2-month-old, 1.6 kg emu was referred to the University Veterinary Clinic of Liège for a permanent acute-onset lameness without associated trauma of the right leg. The patient was unable to maintain a standing position. The rest of the clinical examination was normal. The preoperative heart rate and respiratory rate were 180 bpm and 24 bpm, respectively. No blood work was performed. On clinical examination, there was a periarticular swelling of the soft tissues of the right intertarsal joint, with persistent hyperflexion and limited amplitude movement of this joint, associated with a lateral luxation of the Achilles tendon. The tendon could only be repositioned manually when the intertarsal joint was in hyperextension. Radiographs confirmed the presence of synovitis and periarticular soft-tissue inflammation of the intertarsal joint, with no radiographic bone abnormality. A lateral luxation of the Achilles tendon was diagnosed. Meloxicam at 0.5 mg/kg (Meloxidolor 5 mg/mL, Dechra, Bladel, Nederland) was administered intramuscularly once daily perioperatively [6]. A medial retinaculum reconstruction with a ‘bootlace’-style suture with polyglactin 910 (size 1, Vicryl, Ethicon, Machelen, Belgium) and a lateral reticular release were performed to achieve a complete reduction of the luxation. Ten days later, the same condition developed on the animal’s left leg, with a luxation that was not manually reducible. The same procedure was performed.

For both anaesthetic episodes, premedication consisted of butorphanol 0.1 mg/kg (Butomidor 10 mg/mL, Ecuphar, Oostkamp, Belgium), ketamine 2 mg/kg (Nimatek 100 mg/mL, Dechra, Bladel, Nederland) and midazolam 0.2 mg/kg (Dormazolam 5 mg/mL, Dechra, Bladel, Nederland) administered intramuscularly into the iliotibialis lateralis muscle. Twenty minutes after premedication, anaesthesia was induced with isoflurane (Isoflutek 1000 mg/g, Alvira, Barcelona, Spain) in oxygen via a face mask. The trachea was intubated with a 4.5 mm internal diameter uncuffed endotracheal tube (Curity, Covidien, Bangkok, Thailand). The endotracheal tube was then connected to a non-rebreathing Mapleson F system. Shortly after induction, a 22 G intravascular cannula (IV catheter, Covetrus, Madrid, Spain) was placed in the medial metatarsal vein on the leg opposite to the surgical intervention. Anaesthesia was maintained with isoflurane in oxygen via the endotracheal tube, and physiological parameters were continuously monitored using an integrated multiparameter-monitoring system (VT-9000, Veterinary Technics Int., Ijmuiden, Netherlands). The fresh gas flow was adapted based on the minute volume of the patient to avoid CO_2_ rebreathing. The fraction of inspired isoflurane was continuously adapted with regard to the muscle tone, palpebral reflex and spontaneous movements. Intravenous fluid therapy using a balanced crystalloid solution (Vetivex Ringer lactate solution, Dechra, Bladel, Nederland) was administered at a rate of 10 mL/kg/h, as previously described in birds [7]. The patient received perioperative amoxicillin with clavulanic acid (15 mg/kg, Amoxiclav Sandoz 1000 mg, Sandoz, Vilvoorde, Belgium) intravenously. Respiratory rate, heart rate, lead II ECG, haemoglobin oxygen saturation, end-tidal carbon dioxide (PE’CO2), end-tidal isoflurane concentration (FE’ISO) and body temperature were continuously monitored and recorded every 5 min. Body temperature was evaluated using an oesophagus probe positioned within the coelom. Non-invasive oscillometric arterial pressure was measured with the cuff placed over the tarsus every 3 min. The size of the cuff was selected based on its width, representing 40% of the limb circumference (size 2 for 4–8 cm limb circumference). The emu was positioned in lateral recumbency with the operated leg uppermost and the feathers of the leg plucked out over the thigh to allow for a good ultrasound window.

An ultrasound-guided sciatic nerve block was performed as part of the analgesic protocol using a 22 G, 50 mm Quincke needle (Sonoplex^®^STIM, Pajunk, Geisingen, Germany) and a linear ultrasound probe (L7VET, Clarius, Vancouver, BC, Canada). The relevant anatomical structures and the sciatic nerve were localised using the technique already described in other avian species [2]. The probe was placed on the lateral side of the thigh in a transverse orientation to the long axis of the limb at mid-distance between the hip and the knee (Figure 1). Both components of the sciatic nerve, the common peroneal and tibial nerve, were identified caudal to mid-femur between the lateral and medial muscles of the thigh (iliotibialis lateralis and iliofibularis flexor cruris lateralis on the lateral side, and flexor cruris medialis and femurotibialis internus on the medial side) [8]. The needle was advanced in-plane through the caudal aspect of the thigh and advanced in a caudo-cranial, latero-medial direction, aiming to reach the sciatic nerve proximity (Figure 2). Levobupivacaine 1 mg/kg (Levobupivacaine 0.5 mg/mL, Fresenius Kabi, Huis Ter Heide, Netherlands Nederland) was injected around the nerve through the prefilled needle using a 2 mL volume syringe. Prior to local anaesthetic injection, a negative aspiration test was performed to confirm extravascular needle placement. No resistance to the injection was felt. The sonographic image confirmed the sciatic nerve was embedded in the local anaesthetic solution (Appendix A). Intraoperative nociception was considered when the heart rate and respiratory rate increased sharply by 20% of the baseline values in response to a surgical stimulation. Ketamine 0.5 mg/kg was prepared as a rescue systemic analgesic drug.

General anaesthesia durations were 130 and 150 min (trials 1 and 2). Surgical procedures lasted 40 and 70 min, respectively, with a nerve-block-to-incision time of 35 and 40 min (trials 1 and 2).

Throughout both procedures, FE’ISO was adjusted between 0.9 and 2.1% to limit spontaneous movements. While spontaneously breathing, the bird’s PE’CO_2_ varied between 23 mmHg and 83 mmHg, and subsequently intermittent positive pressure ventilation was started using a “mechanical thumb ventilator” (SAV04, Vetronic, Abbotskerswell, UK), aiming to maintain the PE’CO_2_ under 45 mmHg. Both heart rate and mean arterial pressure (MAP) ranged between 80 to 190 bpm (value at admission 180 bpm) and 50–105 mmHg (mean value before the first surgical intervention was 87 mmHg), respectively. Bradycardia (defined as heart rate under 100 bpm) associated with hypotension (defined as MAP under 60 mmHg) was treated with atropine titrated to effect on one occasion (37.5 μg kg^−1^, Atropine Sulfate, 1 mg mL^−1^, Sterop, Anderlecht, Belgium). All other monitored parameters did not vary significantly during the procedures.

The recovery was uneventful on both occasions. Following surgery, the patient continued to receive meloxicam at a dosage of 0.5 mg/kg intramuscularly once daily. The diet returned to normal shortly after the patient fully recovered from general anaesthesia. Postoperative pain was evaluated using a subjective scoring system (visual analogue scale) by the veterinary healthcare team. No requirement for additional analgesic administration was documented.

## 3. Discussion

General anaesthesia was uneventful. Initially, significant fluctuations in heart rate and mean arterial pressure (MAP) were observed while titrating anaesthetic depth to the optimal level. Notably, these variations were not associated with any surgical stimuli, as they occurred prior to incision. Currently, the minimum alveolar concentration (MAC) for isoflurane in this species remains undetermined. However, the existing literature suggests targeting an inspired isoflurane concentration of 2–3% during nonpainful procedures [9].

We anticipated this surgery to induce significant nociception in this patient. However, due to limited research on avian pain responses to various surgeries, this assessment is based on data from dogs (Monteiro 2022). Given the lack of nociception signs observed during the surgical procedure, it can be inferred that the sciatic nerve block was successful on both occasions. However, due to the need for the bandage immobilization of the emu’s limb to prevent post-anaesthetic injury, an accurate determination of the precise duration of myorelaxation and desensitisation could not be established. Based on an objective assessment of vital signs including heart rate and the subjective reporting of appetite and behavioural changes, no evidence of postoperative pain was identified. Great advantages of the regional anaesthesia, and the sciatic nerve block in particular, have already been proven in other avian species [1,2,10].

The emu’s hind limb is essentially built for power and speed on land. The combination of a lightweight but strong femur, a broad pelvis for muscle attachment, powerful leg muscles, and a three-toed foot for stability showcases how evolution has sculpted this bird for a completely different lifestyle compared to its flying relatives. This may influence the ultrasound view of the mid-femur muscles around the sciatic nerve. Emus have particularly well-developed gastrocnemius and iliotibialis muscles [8]. Detailed research comparing emu hind limb innervation to other birds is scarce, as most studies on avian hind limb innervation focus on flying birds. The basic function of the hind limb muscles is likely to be controlled by similar core-nerve patterns across most birds. However, the emphasis on running in emus compared to flying in other birds suggests there might be subtle variations in the organization and distribution of nerve fibres within the muscles.

A review of the literature identified only one prior report of ultrasound-guided sciatic nerve block in an avian species [2]. This case involved a duck with a tibiotarsal fracture and utilized a combination of sciatic and femoral nerve blockades with lidocaine 2% at a total dose of 8 mg/kg. In our case, the lesion was located distally in the tibia, rendering femoral nerve blockade unnecessary. Levobupivacaine was chosen as the local anaesthetic due to its extended duration of action, potentially providing postoperative analgesia. 

In less common species literature is scarce and we do not take full advantage of the regional anaesthesia technique already described in dogs. Even so, an observational survey study involving a significant portion of respondents (over one-third) revealed that many use adapted regional anaesthesia techniques for pain management in birds [4]. However, what limits the use of these techniques in these species are the lack of published data on the local anaesthetic drugs and regional anaesthetic techniques and limited knowledge of anatomy. The lack of anatomical knowledge may be easily overcome using the ultrasound technique, which allows the practitioner to identify and to evenly distribute the local anaesthetic drug around the targeted nerve. However, to our knowledge, there are no published data on the success rate of the ultrasound-guided mid-femur sciatic nerve block in any animal species.

Ultrasound visualization for sciatic nerve blockade has become a well-established and efficient approach in dogs since research conducted in 2007 [11]. This technique has since become a standard and routine practice for nerve blockade in canine tibial and lower hindlimb procedures. This regional anaesthesia technique carries potential risks, including the inadvertent intravascular injection of the local anaesthetic agent and nerve injury due to intraneural placement or needle-tip transection. Fortunately, the risk of intravascular and intraneural deposition was mitigated by negative aspiration before injection and the absence of resistance to inoculation, respectively. Additionally, a Quincke point needle was employed to minimize the likelihood of nerve transection. The present case report aims to bridge the translational gap by exploring the application of this established technique in an avian species.

The co-administration of butorphanol, ketamine and meloxicam may have synergistically enhanced the analgesic efficacy of the regional anaesthetic technique. This implies that we are unable to exclude the fact that the systemic analgesic drugs themselves might have been sufficient for controlling intraoperative nociception. The meloxicam dosage used in our case may have been insufficient. A pharmacodynamic analysis of meloxicam in emus suggests that the commonly used dose of 0.5 mg/kg once daily may be inadequate [6]. The authors suggest a dose of more than 2.5 mg/kg/day for emus with a body weight under 5 kg.

While this case report demonstrates the feasibility and apparent efficacy of the presented regional anaesthetic technique in this species, a well-designed, randomized controlled trial (RCT) is necessary to definitively evaluate its generalizability and safety profile. The RCT should enrol a sufficient sample size and employ objective outcome measures to assess both the success rate and incidence of adverse events. Only through such rigorous investigation can we determine whether this method exhibits a clinically acceptable success rate with a manageable risk of complications in a larger population.

## 4. Conclusions

This case report demonstrates the feasibility and relative ease of applying an ultrasound-guided sciatic nerve block in an avian species, specifically the emu. However, the analgesic efficacy of the block may have been confounded by the concurrent administration of systemic analgesics.

## Figures and Tables

**Figure 1 animals-14-01178-f001:**
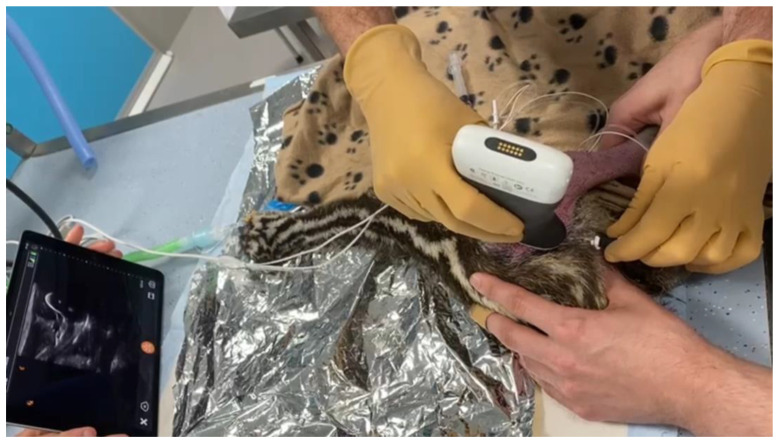
Ultrasound probe positioned over the right pelvic limb of a juvenile emu. The probe was positioned laterally on the thigh, in a transverse plane relative to the longitudinal axis of the limb, at the midpoint between the trochanter major and the femoral condyle. An insulated needle was then advanced percutaneously in a craniomedial direction within the same plane, targeting the vicinity of the sciatic nerve.

**Figure 2 animals-14-01178-f002:**
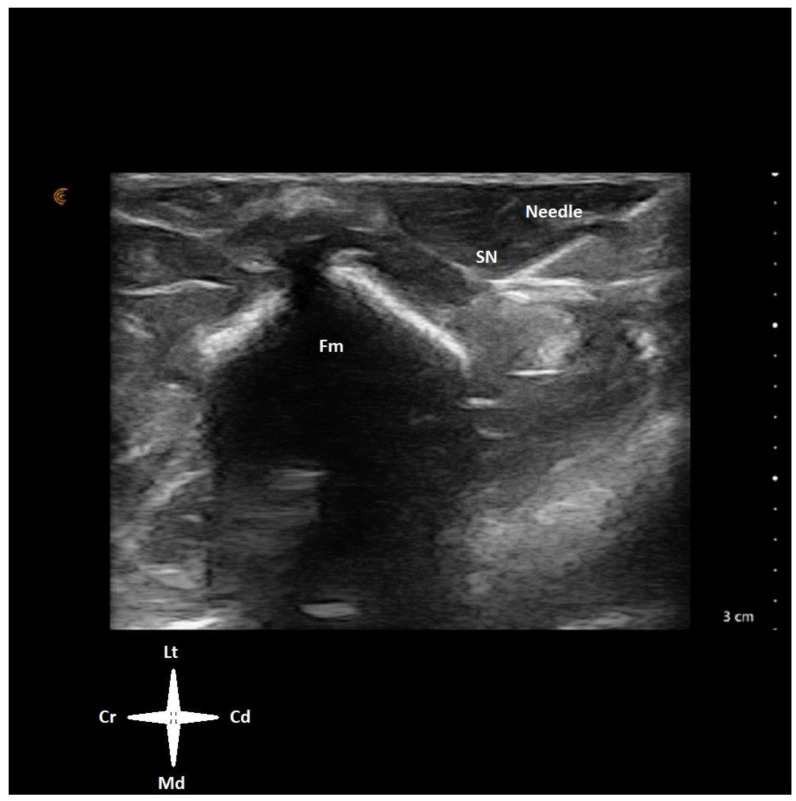
Ultrasound window of the left pelvic limb of a juvenile emu positioned in right lateral recumbency. The transducer was placed on the lateral side of the thigh in a transverse orientation to the long axis of the limb at mid distance between the hip and the knee. An insulated needle was advanced in-plane in a cranio-medial direction aiming for the proximity of the sciatic nerve (SN). Cr—cranial, Cd—caudal, Lt—lateral, Md—medial, Fm—femur.

## Data Availability

No new data were created.

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
