# Peer review of "The Successful Use of an Ultrasound-Guided Mid-Femur Sciatic Nerve Block in a Juvenile Emu (Dromaius novaehollandiae) under General Anaesthesia"

_animals, 2024, doi:10.3390/ani14081178_

Round 1

Reviewer 1 Report

Comments and Suggestions for Authors

Many thanks for your submission, and the time taken to report this original case.

Please find my comments below.

L20: nociception instead of pain

L21: perineurally

L27-30: this feels like you are repeating the abstract, and not introducing the report. I would remove these lines and expand on the background; maybe reporting the other successful cases of avian loco-regional blocks.

L37: it is not really a new method though. Sciatic nerve blocks have been performed in multiple species including avian species. I think that the aim of this case is to report the feasibility of the technique in a different species, and prove the transferability of the technique across species.

L51: I am not certain i would report the immature skeleton here as this is expected for a patient of this age.

L53: "daily" - please be more specific as to whether it was once daily, twice daily, three times daily? Did you extrapolate the dose from another species or is there a reference that supports the use of this drug at this dose, in this species, for this age?

L54: Vicryl is a trade name, so if you really want to be specific regarding the suture material that was used, I believe you should use the specific name of the suture material.

L69: I would suggest that you list the monitoring equipment used before reporting that CO2 rebreathing was observed.

L70: I am certain that there were other signs of anaesthetic depth that you used to adjust the amount of isoflurane delivered - loss of spontaneous ventilation, loss of palpebral reflex for example.

L73: 0.15 mg/kg - are you sure there is no typo for this dose? It seems so much lower than what is used in other species?

L76: deep oesophageal as opposed to superficial oesophageal? What do you mean by deep? Intra-thoracic portion of the oesophagus?

L86: why based on dogs and not other avian species?

L96: did you aspirate prior to injecting to prevent an intravascular injection?

L97: it would have been interesting to combine the U/S technique with a nerve stimulator, to confirm that the structure you observed was indeed carrying motor (and therefore hopefully sensory) information to the distal aspect of the limb.

At that stage of the report, you are not certain yet that what you saw is actually the nerve. Moreover, the US picture provided (Figure 2) labels the sciatic nerve as a hyperechoic structure (possibly hidden by the needle), whereas in my experience of dogs and cats, the sciatic nerve is a double hypoechoic circular structure surrounded by a hyperechoic sheath giving it a "goggles-like" appearance. That is not what is observed in the picture provided, and I wonder what made you feel confident about the fact that the labelled structure was indeed the sciatic nerve.

L140-146: this paragraph could apply to literally any case report. I think it would be more interesting to use this space to expand further on the differences between the blok you are reporting here, the other avian reports of the same technique, and the canine technique.

L148: same comment, I think a conclusion that is more targeted to what you are reporting would be more interesting. This concluson currently feels more like a general introduction, stating why performing blocks in avian species should be considered. The conclusion from this report is that this block seems to be feasible in this species, without any apparent complexity or complication and that levobupivacaine provided appropriate analgesia for the duration of the procedure. I would put a brief comment on the fact that the presence of butorphanol and ketamine in the anaesthetic protocol prevent you from being 100% certain of the block efficacy due to their analgesic properties.

Comments on the Quality of English Language

- Slipped tendon is used everywhere in the text except L47 "lateral luxation of the Achilles tendon". "Slipped tendon" is neither a clinical or scientifically accurate description of the condition, and a minima you must define it early in the text, but I would overall refrain from using this denomination.

L33: changes to patients

L33-34: For this reason [...]by using perioperative systemic analgesics and regional anaesthetic techniques.

L46: "with limited amplitude of this joint remaining in hyperflexion". This is not very clear - do you mean that it was challenging to create hyperflexion of the joint, or that there was a persistent hyperflexion?

L68: "by" -> via

L69: volume minute -> minute volume

L116: "installed" -> started

L127: "temporally" - I think you can remove this word, the rest of the sentence is clear enough without needing it.

Author Response

Dear reviewer,

Thank you for accepting to review our manuscript.

Please find my comments below.

L20: nociception instead of pain

- text was changed

L21: perineurally

- corrected

L27-30: this feels like you are repeating the abstract, and not introducing the report. I would remove these lines and expand on the background; maybe reporting the other successful cases of avian loco-regional blocks.

- text was adapted accordingly

L37: it is not really a new method though. Sciatic nerve blocks have been performed in multiple species including avian species. I think that the aim of this case is to report the feasibility of the technique in a different species, and prove the transferability of the technique across species.

- good point; text was adapted accordingly

L51: I am not certain i would report the immature skeleton here as this is expected for a patient of this age.

- mention removed

L53: "daily" - please be more specific as to whether it was once daily, twice daily, three times daily? Did you extrapolate the dose from another species or is there a reference that supports the use of this drug at this dose, in this species, for this age?

- once daily; more information was added

L54: Vicryl is a trade name, so if you really want to be specific regarding the suture material that was used, I believe you should use the specific name of the suture material.

- the name of the suture material was added

L69: I would suggest that you list the monitoring equipment used before reporting that CO2 rebreathing was observed.

- done

L70: I am certain that there were other signs of anaesthetic depth that you used to adjust the amount of isoflurane delivered - loss of spontaneous ventilation, loss of palpebral reflex for example.

- true, there were palpebral reflex an spontaneous movement. However, increase in muscular tonus was always the firs sign of superficial depth of anaesthesia. Text was supplemented

L73: 0.15 mg/kg - are you sure there is no typo for this dose? It seems so much lower than what is used in other species?

- it was a typo error; the dose use was 15mg/kg; error was corrected

L76: deep oesophageal as opposed to superficial oesophageal? What do you mean by deep? Intra-thoracic portion of the oesophagus?

- mor details were given in the text

L86: why based on dogs and not other avian species?

- good point; text was changed and avian references given

L96: did you aspirate prior to injecting to prevent an intravascular injection?

- yes; info added in the text

L97: it would have been interesting to combine the U/S technique with a nerve stimulator, to confirm that the structure you observed was indeed carrying motor (and therefore hopefully sensory) information to the distal aspect of the limb.

unfortunately the nerve stimulator was not available

At that stage of the report, you are not certain yet that what you saw is actually the nerve. Moreover, the US picture provided (Figure 2) labels the sciatic nerve as a hyperechoic structure (possibly hidden by the needle), whereas in my experience of dogs and cats, the sciatic nerve is a double hypoechoic circular structure surrounded by a hyperechoic sheath giving it a "goggles-like" appearance. That is not what is observed in the picture provided, and I wonder what made you feel confident about the fact that the labelled structure was indeed the sciatic nerve.

  • both the tibial and fibular branches were seen (as in dogs); However, as the needle was getting closer, the exact googles-like aspect was deformed. We double checked with the video we took when the block was performed. I would like to add the video if possible.

L140-146: this paragraph could apply to literally any case report. I think it would be more interesting to use this space to expand further on the differences between the blok you are reporting here, the other avian reports of the same technique, and the canine technique.

- Good point; the text was change and comparisons with other avian species and dogs were added

L148: same comment, I think a conclusion that is more targeted to what you are reporting would be more interesting. This concluson currently feels more like a general introduction, stating why performing blocks in avian species should be considered. The conclusion from this report is that this block seems to be feasible in this species, without any apparent complexity or complication and that levobupivacaine provided appropriate analgesia for the duration of the procedure. I would put a brief comment on the fact that the presence of butorphanol and ketamine in the anaesthetic protocol prevent you from being 100% certain of the block efficacy due to their analgesic properties.

  • the conclusions were change to meet these relevant remarks

All the comments on the quality of English Language were addressed and corrected

Reviewer 2 Report

Comments and Suggestions for Authors

Dear authors,

thank you very much for submission of your case report about sciatic nerve block in an anaesthetised Emu. The case is very interesting and has the potential to show how easily various techniques can be transferred between species. However, the importance of the case report (value for the reader) is missing and also a proper discussion of the case (eg potential complications of this technique etc). As i do understand that the focus is not on the description of the anaesthetic event itself i would still add some more informations about the anaesthesia itself. The images are relevant and showing what has been done, however, a discussion of relevant anatomy is missing. 

some more comments below

Abstract: please include whether the block was successful

title: it would be good to mention that the block was used during anaesthesia, please rephrase the title

- please start introduction with the background and not with the aim/content of your case report

- Introduction very short, please introduce the reader into advantages/disadvantage of combination of general anaesthesia and local blocks.

please also include why your case is interesting

case report: please include at the beginning which leg was initially affected

please include duration of anaesthesia and surgery

HR range of 80-190 is a wide range. please clarify what has been baseline? same with MAP

did breathing varied during the procedure? could this have been a pain response, too?

how long after block did the surgery start

how was recovery? how was weight bearing of the affected leg? 

please include post-operative care, pain etc

Discussion: too short 

please discuss: advantages/disadvantes of this block in this individual, compare anatomy to other species

compare success of this block in various species with your successrate

why bupivacaine chosen? what level of pain is associated with this procedure? pain level post-operative?

other options than ultrasound guided block

discuss shortly your anaesthetic protocol and whether or not it did contribute to intraoperative pain relief

no need to mention pain assessment in birds in your conclusion as you never describe and discuss pain assessment in your article

p[lease include castineiras 2021 for discussion of metacam doses in emu

please also discuss the use of locoregional anaesthesia techniques in this/exotic species (eg Quesada 2022)- discuss translation from one species to another

please also discuss innervation of hind legs in emus - similar or different from other species 

Author Response

Dear reviewer,

Thank you for accepting to review our manuscript and for the good suggestions which will improve it.

Please find below our answers to your suggestions: 

title: it would be good to mention that the block was used during anaesthesia, please rephrase the title

- title was rephrased to mention general anaesthesia was involved 

- please start introduction with the background and not with the aim/content of your case report

- the introduction section was re-organized

- Introduction very short, please introduce the reader into advantages/disadvantage of combination of general anaesthesia and local blocks.

- introduction section got improved with the side effects produced by nociception and the advantages of the association of systemic analgesics and regional anaesthetic techniques

please also include why your case is interesting

- good point! the text was reformulated

case report: please include at the beginning which leg was initially affected

- the information was added

please include duration of anaesthesia and surgery

- information added

HR range of 80-190 is a wide range. please clarify what has been baseline? same with MAP

- the baseline values were added

did breathing varied during the procedure? could this have been a pain response, too?

- the patient was under mechanical ventilation during the surgery as he was in hypoventilation 

how long after block did the surgery start

- data was added

how was recovery? how was weight bearing of the affected leg? 

  • the recovery was uneventful as mentioned in the text
  • the patient was unable to maintain standing position

please include post-operative care, pain etc

- more information regarding the recovery period was mentioned

Discussion: too short 

please discuss: advantages/disadvantages of this block in this individual, compare anatomy to other species

- advantages and disadvantages/risks and anatomical specificities were discussed

compare success of this block in various species with your success rate

- to our knowledge, the success rate of this regional anaesthetic technique is not yet described. We expressed this in the text.

why bupivacaine chosen? what level of pain is associated with this procedure? pain level post-operative?

- the answers for all the three questions are now added in the text

other options than ultrasound guided block

- nerve stimulator guided technique is now mentioned

discuss shortly your anaesthetic protocol and whether or not it did contribute to intraoperative pain relief

- text was added to discuss the effects of the systemic analgesic properties of this anaesthetic protocol

no need to mention pain assessment in birds in your conclusion as you never describe and discuss pain assessment in your article

- text was excluded

please include castineiras 2021 for discussion of metacam doses in emu

- reference is used

please also discuss the use of locoregional anaesthesia techniques in this/exotic species (eg Quesada 2022)- discuss translation from one species to another

- reference and discussion added

please also discuss innervation of hind legs in emus - similar or different from other species 

- a short anatomical discussion was added

Reviewer 3 Report

Comments and Suggestions for Authors

Dear authors,

I read the case report with great interest.

I think it is well conducted, I have very few observations that I hope will help to implement the quality of your report. Thank you very much.

Line 12 and 23: Given the concentration of levobupivacaine used, I would advise you to already include the concept of post-operative pain management in line 12. 

In the end it is a concept that you take up in line 23 and in your discussions.

Line 31: I would put loco-regional analgesia technique.

For the description of the technique you use a dog reference, consider whether to include the following reference given the approach you use

Trujanovic R, Otero PE, Larenza-Menzies MP. Ultrasound- and nerve stimulation-guided femoral and sciatic nerve block in a duck (Anas platyrhynchos) undergoing surgical fixation of a tibiotarsal fracture. Vet Anaesth Analg. 2021 Mar;48(2):277-278. doi: 10.1016/j.vaa.2020.11.001. Epub 2020 Dec 3. PMID: 33563535.

Congratulations

Author Response

Dear reviewer,

Thank you for accepting to review this manuscript.

Please find below my comments

Line 12 and 23: Given the concentration of levobupivacaine used, I would advise you to already include the concept of post-operative pain management in line 12. 

- true the drug and concentration provided the benefit of postoperative analgesia too; text was added 

Line 31: I would put loco-regional analgesia technique.

- text changed

For the description of the technique you use a dog reference, consider whether to include the following reference given the approach you use

  • the discussion section was improved with a paragraph were the current case report is compared with the one written by Trujanovic et al.

Round 2

Reviewer 2 Report

Comments and Suggestions for Authors

Thank you very much for your revised manuscript. The case report is now easy to follow and description and conclusions are sound